# Sustainable Digital Change: The Case of a Municipality

Svala Gudmundsdottir [1,*], Throstur Olaf Sigurjonsson [1,2], Eva Marín Hlynsdottir [3],
Solrun Dia Fridriksdottir [4] and Inga Sol Ingibjargardottir [4]

1   School of Business, University of Iceland, 102 Reykjavík, Iceland; olaf@hi.is
2   Department of Accounting, Copenhagen Business School, 2000 Frederiksberg, Denmark
3   Faculty of Political Science, University of Iceland, 102 Reykjavík, Iceland; evamarin@hi.is
4   Independent Researcher, 101 Reykjavík, Iceland
*   Correspondence: svala@hi.is

**Abstract:** This study concentrates on how change can be effectively managed within the public sector to enhance sustainability. Public institutions are vital in promoting societal well-being and must be capable of adapting to changing circumstances while embracing sustainable practices. The study discusses the importance of digital development and inclusion in the public sector, highlighting the need for organizations to adapt to a changing world and prioritize information technology and user needs. The research methodology involves qualitative research, including semi-structured interviews with employees from the city of Reykjavík, Iceland. The findings emphasize the importance of leaders and middle managers being on board and taking ownership of the digital transformation process. The study also highlights the significance of sustainability in resource management and the innovation in service provision that comes with digital transformation. Overall, the study contributes to understanding change management and digital development in the public sector and provides insights for organizations that are seeking to promote sustainability and adapt to digital advancements.

**Keywords:** change management; sustainability; digital transformation; societal level; leadership

## 1. Introduction

The aim for sustainable development at a societal level means that societies and organizations within them need to be able to adapt to change. Moreover, rapid urbanization combined with the transformation of modern technology enhances this need [1]. Urban sustainability is a highly complex concept with several, often conflicting definitions. As Verma et al. [2] argue, it includes three dimensions, which are environmental, social, and economic sustainability; of these, social sustainability is the least studied. However, they argue that all three dimensions must be explored and examined if the aim is to achieve sustainable long-term quality of life for all citizens. This is essentially the aim of the UN's 17 sustainable development goals [2]. Of these goals, no. 9 (industry, innovation, and infrastructure) [3], 11 (sustainable cities and communities) [4], and 12 (responsible consumption and production) are of particular interest to urban sustainability [1,5]. In relation to this, information and communication technologies are of great importance [5,6] as technology inevitably affects the operating environment of public organizations such as cities. To keep up with rapid technological changes, state and local government organizations in Iceland in recent years have been working to modernize the working environment and transfer an increasingly large part of their services to a digital format [7,8].

The city of Reykjavík has a strategy coined as the Green Plan, which began in 2020 and will continue until 2030. It is the city's long-term plan for finances and investment with the aim of increasing the sustainable well-being of the city's residents. It is intended to outline the city's future vision and combine critical policies and plans behind that vision for the good of society. The Green Plan has many focus areas, including environmental and

climate issues, public finances, and social issues, and all operations should be based on sustainable indicators [9].

The digital transformation of Reykjavík's services is among the more significant projects within the Green Plan. It is an effort to streamline and modernize the city's services and make them more environmentally, economically, and most importantly, socially sustainable. The main changes include adapting the city's services to digital solutions and increasing the proportion of services that can be accessed and provided electronically. These are sustainability efforts that are significantly affected by the change management practices of the city. How the city manages these changes will impact the sustainability initiatives of the city's Green Plan.

In 2019, the city of Reykjavík's Service and Innovation Division was established. With the establishment of the department, the formal development of digital skills within the city was set in motion. Digital professionals and IT teams began working on various projects and activities in the city. Accumulated knowledge emerged in the field, such as data analysis and artificial intelligence, when a special data team was established, and the data manager position was created. In 2022, the city and five other cities worldwide were chosen to participate in the project "Build Back Better", which is intended to speed up the process of digital transformation in order to improve the lives of city dwellers and their quality of life, following the COVID-19 pandemic. The city of Reykjavík received a three-year grant of nearly 2.2 million US dollars to achieve these changes. These funds were utilized to finance the so-called "I-team". The I-team used the data and digital technology to improve public services and find creative ways to create value for society [10]. To better meet the needs and wishes of its residents, the city of Reykjavík has put considerable effort and ambition into improving and simplifying its services and service processes. The city aims to bring the services closer to residents by offering online application platforms and user-oriented services, so that individuals can request services online and monitor the status of their cases at all levels. User-oriented and accessible services increase well-being and ensure equal opportunities for residents, where applicants are not grouped into specific groups. Hence, solutions are devised from a user-oriented perspective based on each person and their needs. The city of Reykjavík's first project in its digital transformation of essential services was the digitalization of financial assistance, where a solution was created that would provide a choice for residents to apply for financial aid online. This project was awarded the Web System of the Year in 2019 at the Icelandic Web Awards. Among other accolades, it was also mentioned in the project review that the site involved a great deal of innovation that was accessible and that the entire process was user-friendly [11].

The announced changes encompass a broad range of modifications that significantly impact both the services provided by the city and the working environment for its employees [11]. Transitioning towards digitalization presents numerous challenges for both employees and stakeholders [12] due to the fact that it permeates nearly every aspect of human activity and has widespread effects on the economy and contemporary society [13]. The underlying significance of establishing a socially sustainable foundation for delivering well-being and exemplary services in order to promote social equity within society amplifies the gravity of these challenges. Consequently, this study addresses the research question of which factors contribute to the successful implementation of sustainability efforts in the city of Reykjavík during a transformation towards digitalization on a large scale. The paper is organized in a manner that first provides a comprehensive review of the relevant literature, followed by a chapter detailing the methodology employed. Subsequently, the results are presented, accompanied by a discussion and concluding remarks.

## 2. Literature Review

### 2.1. Managing Change within the Public Sector to Promote Sustainability

Public institutions play a crucial role in promoting societal well-being and, therefore, must ensure their sustainability in operations. Achieving sustainability requires adapting to a changing environment and incorporating sustainable practices into the core values

and operations of the institution. This necessitates the adoption of new technologies and innovative practices, which can be challenging to navigate without effective change management [3,4,14]. The digital entrepreneur Graeme Wood (ed.) once said, "Change has never been faster than it is now, and it will never be slower again" [15]. This quote is relevant today as every day brings innovations and changes that organizations and society must adapt to. When organizations decide to make changes, this must be conducted in an organized manner. It often seems to be the case that changes are heavy and difficult to implement within public administration. Although a lot of time is spent planning modifications, they cannot be delivered properly and thus become entrenched [4]. Or, because they are so time-consuming to prepare, their purpose is almost forgotten when the changes are eventually implemented [16]. Now that technological innovations are constantly developing and new possibilities emerge in operational form, public institutions and administration must not lag behind at all [1]. Therefore, companies and institutions must spot opportunities to improve existing or rather develop new services or products, thus preventing stagnation. When it comes to change management within public administration, an assessment and analysis of the available theoretical data was carried out by Fernandez and Rainey [4]. Their article that was published in the Public Administrations Review analyzed the research and theories on organizational change in the public administration. During this analysis, it became clear that many conflicting theories have been presented, and that research results often contradict each other. Although there are usually clearly visible differences between the theories and models that have been proposed regarding change management, there are also many similar aspects, and there is a certain consensus among scholars that specific steps must be taken in order to increase the likelihood of the successful implementation of changes. The study by Seijt and Gandz [17] uses Kotter's eight-step model to assess the importance of leadership skills in a change process. They argue that Kotter's eight-step model is the starting point for most managers, directors, entrepreneurs, consultants, and others who are tasked with leading change or helping others do so. Their results showed that when leaders are perceived as having honesty, humility, kindness, self-discipline, fairness, and cooperation, those who might oppose are more likely to be motivated to support ongoing change and will, therefore, feel comfortable in the change process and be open to further changes. On the other hand, if managers are perceived as dishonest, deceptive, unfair, arrogant, rude, and lacking empathy for those who have difficulty with change, they are more likely to create hostility and resistance to future changes, even if the current changes may prove to be successful. Kanter [18] further argued that the most important qualities managers and leaders can bring to the change process are passion, conviction, and trust in others.

Dumas and Beinecke [19] examined the field of change leadership at the beginning of the 21st century. They discuss what is needed to bring about effective changes within organizations. There is much in common between their ideas and what has been discussed above, both in the model of Kotter [20] and the proposals of Fernandez and Rainey [4]. Dumas and Beinecke [19] believe that successful change requires the widespread recognition of the need for change, the participation of many stakeholders in the process, effective communication about the need for change, support from top management, and political support. They also state that in changes, there must be a mix of management that comes from above (top-down) and below (bottom-up), which empowers employees to act with responsibility and participation, and enables and builds commitment to the changes. The role of managers has been considered necessary in the change process [21]. That said, managers can primarily find it challenging to deal with contradictions and tensions due to changes from above and pressure from the staff below. Their role is, i.e., occupying a central position in the organizational chart, where they are responsible for implementing the policy of senior management by ensuring that lower-ranking employees fulfill their roles [22].

Policies are formulated at the top but implemented from the bottom up. Therefore, communication is critical for implementing a new work strategy and achieving the desired

results. Studies on the role of communication during organizational change focus on issues such as the design and adoption of the change process, the creation of employee participation [23], and the role of management communication [24]. By [25] has emphasized the importance of everyone moving in the same direction and it is further argued, when looking at organizational development, deployment, and success, pitting employees and managers against each other does not encourage a team approach or a sense of shared responsibility. Rather, it creates a gap between them and leads to an inability to fulfill the main potential of an organization.

*2.2. Digital Development*

Digital inclusion is a term that essentially refers to the implementation and significant use of digital applications for social and economic benefit by considering the quality of computer equipment, ownership, digital literacy, and digital skills [26]. Much of what we know in societies and economies today is moving to digital form, which increases the need for a deeper understanding of the associated changes. With continued digital development, software becomes increasingly complex, and this can be a challenge for different groups of people, making it difficult for some to be active participants in the economy and society of the 21st century [27]. Digital participation can propose challenges, for example, the differences between younger and older generations. Digital parity refers to the idea that different social and economic outcomes between groups, e.g., in urban or rural areas, can partly be attributed to differences in internet access, levels of digital skills, and digital mindset. All groups must adapt to digital solutions so that societies can sustainably thrive in this evolving digital age [28].

Bridges and Bridges (2016) assert that organizations must effectively respond to the dynamic nature of the contemporary world. Recent years have witnessed significant transformations in companies and organizations, with technology being pivotal in reshaping the operational landscape. The pursuit of sustainability necessitates the adoption of novel technologies and innovative practices. In this context, change management emerges as a crucial tool for institutions to effectively navigate the challenges associated with incorporating these new tools and methodologies. The emergence of startups as disruptive forces in established industries, coupled with the revolutionary impact of the internet, social media, and applications on communication, has fundamentally altered the fabric of our society [6]. The advent of instantaneous and continuous information dissemination has profound implications for governments, municipalities, and communities alike. Organizations that fail to keep pace with technological advancements will inevitably lag behind, while those that resist adaptation will likely face obsolescence [29]. Implementing sustainability measures often requires a fundamental shift in organizational culture and mindset. Change management methodologies offer valuable support in facilitating this transformation by actively involving employees, aligning organizational values, and fostering a sense of ownership and commitment towards sustainability goals, such as the Sustainable Development Goals (SDGs) [30].

In times of change and technological progress, it is therefore essential that the government prioritizes the use of information technology and the needs of its users, as to ensure that the public sector does not fall behind in a technological world [31]. The Association of Icelandic Municipalities works on cooperation between municipalities in digital development, focusing on collaboration with local authorities, Reykjavík, and the state [8]. Iceland's strategy for the digital services of the public sector aims to increase competitiveness, make the infrastructure safer, improve public services, and make the city of Reykjavík, like the state and other local authorities, undergo a digital transformation, which is intended to adapt the services that the city provides to digital solutions [11]. The digital development of the city is designed to revolutionize how services are provided to residents and how jobs are completed [7,11]. The project is extensive and covers all establishments and staff in the city of Reykjavík. The city's focus on digital transformation is set up in four elements. Firstly, to modernize systems, the goal is to simplify the organization of systems and pro-

cesses and increase efficiency. Secondly, the focus is on the innovation in ways to deal with challenges in the city's operations. Automated administration is the third priority, thus facilitating application processes and access to services. Fourth is using data that the city can use to identify opportunities for the optimization and value creation of the modern environment [7].

## 3. Materials and Methods

The study employed a qualitative research approach, utilizing sixteen semi-structured face-to-face interviews with durations ranging from 33 to 55 min. Purposeful sampling was adopted as the methodological strategy, as this allowed the researchers to obtain valuable insights into a specific topic by selecting participants who could provide relevant information on the subject matter. Additionally, snowball sampling was employed, initially identifying a few participants who met the predetermined criteria for participation. Subsequently, the researcher obtained references from these initial interviewees to identify additional potential participants for the study [32]. In this study, the recruitment of participants was facilitated through the utilization of a snowball method and a purposive sampling method. The selected participants were individuals employed by the city of Reykjavík, specifically in the development and innovation or welfare departments. These departments were specifically chosen due to the development and innovation department's responsibility for overseeing the city's digital transformation implementation, the welfare department's notable progress in the transformation process, and their proximity to end users.

The subsequent data analysis involved preparing and organizing the collected data for further examination. Common themes within the data were identified through a process of coding. Finally, the gathered results were comprehensively described [33]. It is important to note that this research was conducted without a pre-existing theory, and instead adopted a theory approach that was formulated based on the data generated and acquired throughout the research process [32,34]

Table 1 displays the interviewees' composition, comprising managers and other employees who had participated in digital transformation projects. The gender distribution among the interviewees consisted of nine men and seven women. The age range of the interviewees spanned from 34 to 50 years, with their tenure at the city of Reykjavík varying from 1 to 17 years. The interviews were conducted between the spring of 2022 and 2023. To ensure confidentiality, each interviewee was assigned a numerical identifier from one to sixteen. All interviews were recorded and transcribed verbatim. These transcriptions were thoroughly examined, and prominent themes were identified [32,34]. The researchers had no affiliation with the activities of the city of Reykjavík before commencing their research, and they did not have any personal acquaintances among the interviewees. It is essential to acknowledge that some of the researchers reside in the city of Reykjavik and have direct exposure to the city's services. This exposure might inadvertently influence their perspectives on digital development in that locality. Nevertheless, the researchers diligently held a neutral stance and abstained from expressing personal opinions during the interview.

**Table 1.** Sample demographic (*N* = 16).

| Participant | Work Experience in Years within the City of Reykjavik | Position | Department | Gender | Age |
|---|---|---|---|---|---|
| P1 | 6 | Manager | WD | M | 34 |
| P2 | 1.5 | Employee | DID | M | 50 |
| P3 | 1 | Manager | DID | M | 39 |
| P4 | 2 | Employee | DID | F | 45 |
| P5 | 15 | Employee | WD | F | 41 |
| P6 | 15 | Manager | WD | F | 44 |

| Participant | Work Experience in Years within the City of Reykjavik | Position | Department | Gender | Age |
|---|---|---|---|---|---|
| P7 | 1.5 | Employee | DID | M | 44 |
| P8 | 2 | Manager | DID | M | 42 |
| P9 | 17 | Manager | WD | F | 39 |
| P10 | 6 | Manager | WD | M | 34 |
| P11 | 1.5 | Employee | DID | M | 50 |
| P12 | 2 | Employee | DID | F | 45 |
| P13 | 15 | Employee | WD | F | 41 |
| P14 | 15 | Employee | DID | F | 44 |
| P15 | 1.5 | Manager | DID | M | 44 |
| P16 | 2 | Manager | WD | M | 42 |

## 4. Results

### 4.1. Change and Managing Change

The interviews were initiated by discussing the process of change and the interviewees' experience of the city of Reykjavík's strategy in "digital travel". The main changes have already been implemented and the visions for the future were also discussed. An additional focus was on the interviewees' experience of going through a change process within the public sector. Most interviewees mentioned that a "time of change" can be difficult. Interviewee 6 said the process is about "all kinds of prioritization", and even structural changes might be implemented simultaneously, "then maybe it's such a pain while it's going on because then you're both working within the old track, at the same time you're implementing a new one and sometimes it is a long way until you have reached the end of the road". Interviewee 8 echoed the same sentiment and mentioned that employees need to be patient during a period of change:

> *Things take longer while people are conducting something new to them. After that initial work has been undertaken, things take less time. But employees need to be patient during that first phase.*

Several interviewees expressed the opinion that more emphasis should be placed on decommissioning old methods alongside introducing new ones, and the importance of using the older ones, which are intended to be replaced, alongside the new ones. Interviewee 4 said it can be complicated when using an old and a new system simultaneously and expressed it in the way that "suddenly you start using both solutions, and you have the choice to continue using the old solution". Interviewee 1 said he wanted a "clearer architecture" around a system where one route would be chosen and "run on", instead of using two systems with the same functionality. Several interviewees talked about so-called legacy systems. Interviewee 8 explained what he meant by such a system in the following way: "When we talk about legacy systems, it is such an old system that is not maintained". Interviewee 4 said that it went well when it was decided to turn off the old system as soon as the new one was implemented: "It would just be read-only, so people are forced to welcome the changes". Interviewee 5 also described the challenges staff face when replacing old systems, saying:

> *They are working very hard to eliminate old legacy systems, but it is tough and challenging when you have staff who have worked for a long time and are adapting to old systems; it takes so much work that it trusts what is new [...] if you are a senior employee who has just been working on the same things even on paper all the time. Then, you are going to transform it. You must trust this process, and it's just a big challenge in many areas.*

Interviewee 4 also said that getting staff to use new systems could be difficult and mentioned that much money could be spent on developing and designing a new system. However, if no one wants to use that system, it would simply be a waste of money by its very nature. Several interviewees spoke of the goal of digital transformation as simplifying things. In that context, the interviewees talked about the possibility of saving money, e.g., by shortening processes. Interviewee 3 considered the benefits of digital transformation to be increased quality and greater efficiency and added: "I am sure that there will be financial benefits, even if it has not started to pay off much yet". The interviewees agreed that the Reykjavík City Welfare Division staff would be well prepared for the changes. Respondent 5 described, in this context, i.e., the implementation of the Workplace application, a communication as well: "It [the implementation] was perhaps the most successful in the welfare sector because everyone was just so ready". Interviewee 4 agreed and noted that the staff in the welfare sector had been very productive in the digital journey. Several interviewees also said the welfare sector has been active in digital transformation. Interviewee 7 mentioned that the welfare sector provided a lot of direct service to users, and, therefore, there was an "urgent need to jump on this bandwagon".

However, several interviewees also mentioned obstacles the welfare sector had encountered in the digital transformation. Some interviewees expressed that the field was vast, with different categories of issues and many separate establishments. Interviewee 6 mentioned, for example: "We have 3300 employees in various jobs". Interviewee 1 added that it is "no joke trying to lead digital transformation for a huge platform". Interviewee 1 also mentioned that the department's staff have all levels of education, from having minimum education to being highly educated. Interviewee 6 also said that in some departments, employees do not work on a computer and do not open e-mail or the platform Workplace where information is shared. Interviewee 6 added, in that context, "I think that if you ask employees who work at a housing center for people with disabilities, I'm not sure that they would connect with this digital journey". Almost all interviewees talked about the importance of "breaking down the silos". Interviewee 5 considered breaking down these organizational chart walls to be one of the main benefits of digital transformation and said:

> As soon as we make things more transparent. It's easier to access information; we can customize the information needed ... we can also begin to encourage people to show initiative and find ways to work together.

The interviewees agreed that they learned a lot from the change process, and interviewee 6 articulated it effectively when he said: "All change management and everything new like that, a thousand mistakes are made, but we also learn from them, and they are so important to get us forward". Interviewee 4 agreed: "We are not afraid to make mistakes and change as we go". Most interviewees touched on how it is to go through a change process within the laws and rules that apply to public operations. Interviewee 3 said that general operations are subject to certain obligations that set limits and added that the entire operation of the city was very firmly established. Jobs were defined according to old practices. Processes had to be reviewed and redefined for new challenges, and job descriptions had to be updated. Respondent 7 mentioned in this context that:

> For example, we have lawyers who are active participants in our processes because, you know, everything we are dealing with is naturally at an administrative level, which means that we have to follow the law in one way or another in all of our application processes, publication, or answers.

Most interviewees touched on how politicians and other interested parties influence changes in the public sector; interviewee 3 commented, "Politics puts this at the forefront and places a lot of resources and effort (power) into this topic". Interviewee 3 added that, in public operations, the expectations, priorities, and political priorities must be met, and said that it is not only possible to manage the changes "based on what the management thinks and based on what we see that the users are calling for". Interviewee 2 said on a similar note:

*"Of course, it's nice to see how politics greatly influences such priorities, but here it's just how it is: A normal thing". Digital development is taking place in society as a whole".*

Several interviewees touched on a particular debt in the city's technological development, and interviewee 3 said, "Naturally, there is a certain amount of technical debt going on at the City of Reykjavík". Interviewee 2 said there is a great need for digital transformation in the city of Reykjavík and other municipalities. Interviewee 6 then spoke about the city's staff needing digital transformation:

*People feel the systems are outdated. I can find someone who thinks it's all ridiculous and pointless. Still, I think the whole reason the welfare sector is jumping on the bandwagon [...] is the need, and everyone sees the need, and we want to get somewhere else; this is meeting people of the day daily, somehow obstacles in systems, obstacles in applications, so I think that people see the opportunities.*

Interviewee 6 added that digital transformation is a process that takes time, and if it is to be carried out well, it cannot be rushed through: "We are in data debt just really on all fronts as a city, so it's all so much work". The interviewees agreed that there was an ambition to improve the city's technical debt and advance in digital administration.

*4.2. The Gatekeepers*

Participants all expressed the importance of leaders and/or middle managers being on board and being prepared to lead their group forward in the digitalization journey. The importance of employees believing in the project and being dedicated to the project is clear. This theme emphasized how managers need to take ownership and possess the skill to communicate the vision. Interviewee 5 touched on the challenge of leading a digital transformation process in a workplace as large as Reykjavík and said "We have a lot of departments and managers, and we need to try somehow to create some kind of journey and get people onboard". Interviewee 1 further said that "People have various backgrounds, and trying to get everyone to travel in the same direction just needs solid leadership". It was further emphasized that multidisciplinary collaboration is essential early in the process, where trust is critical. Communicating the vision and benefits of the change initiative to employees and managers needs to start earlier in the process to increase understanding and buy-in, particularly in such large change projects that are continuously ongoing and constant. Interviewee 5 believed that the success of the digital transformation in the welfare sector could be attributed to the fact that there was a manager who had confidence in the staff and said that it also helped that the manager in question had training in change management.

Several interviewees spoke of the success of having team members referred to as "digital leaders", one digital leader belonging to each department at Reykjavík City. They form a team to ensure an overview of all areas' challenges and support the overall vision. The digital leaders are responsible for working with management and employees, e.g., by identifying opportunities and mapping the status of services. Interviewees 10 and 13 confirmed this need for interdisciplinary teams, and interviewee 3 said "The role of the digital leaders is to be a bridge". This was further explained as, among other things, harnessing the power of the fields and mapping the needs for a digital journey. However, some interviewees also discussed information overload and that care must be taken when sending information to staff. "The City of Reykjavík is a workplace of 10,000 people. We try to use innovative and multiple methods to introduce ideas. . .. we know people are busy, so we can send out the information [to staff], but we do not know if they read it. We make sure the information is accessible. . ." (interviewee 13). This also has to do with what to expect. Big projects like digital transformation need to be organized stepwise, as pointed out by interviewee 12: "We need to be clear in managing expectations, helping people to see that we will finish the elephant, one bite at a time". In this relation, complying with and abiding by the law is essential, as the interviewees said that public services and governance need to comply with the law. New processes are susceptible and can propose several challenges that can result in complaints from citizens or lawsuits. Interviewee number 9 said:

*We have lawyers on our teams because many things we do or say touch on a governing level, which means we need to obey the law in one way or another, for example, in what we write, answer, or publish.*

### 4.3. Digital Transformation for Improved Resource Efficiency

Many of the interviewees highlighted that digital transformation goes beyond operational improvements and further contributes to sustainable practices. The recurring theme was the reduction of waste through digital processes. Interviewee 1 emphasized the importance of reducing waste while ensuring that it does not compromise service quality, emphasizing the need for digital transformation to support sustainability goals. Interviewee 6 specifically mentioned the benefits of electronic applications, stating that users no longer need to travel between places to apply for services, resulting in reduced carbon emissions and resource consumption. Respondent 4 further emphasized the environmental advantages of digital communication, noting that data can be transmitted online instead of relying on printing and physical transportation. In addition to reducing paper waste, these digital processes also contribute to space-saving, as highlighted by Interviewee 4, who acknowledged the benefits of not storing printed data in physical folders that occupy significant physical storage space. The discussion on sustainability went beyond waste reduction and extended to overall environmental impact. Interviewees acknowledged that digital transformation enhances sustainable work practices through increased data security and the reduction of physical storage needs. By minimizing the reliance on printed materials and physical infrastructure, organizations can mitigate their carbon footprint and contribute to a greener future. These advancements align with contemporary technological developments and societal demands for sustainable practices. Interviewee 6 emphasized that a users' familiarity with online transactions in other areas of life, such as online banking and shopping, has built expectations for digital access to essential services. This indicates that society as a whole is moving towards a more sustainable and digitally driven future. Moreover, interviewees recognized that digital transformation enables more efficient work processes, eliminating redundant tasks and promoting overall productivity. Interviewee 5 mentioned the importance of adopting a "solution-oriented mindset" and optimizing work practices, which not only improves efficiency but also contributes to sustainability efforts. Streamlining operations and reducing unnecessary paperwork can result in resource savings and contribute to a more sustainable workplace.

Considering these insights, it is evident that digital transformation and sustainability are intricately linked. By embracing digital technologies and practices, organizations can reduce waste, minimize the environmental impact of their operations, and improve overall sustainability. The positive outlook on the digital journey shared by interviewees reflects their recognition of the benefits of sustainable practices and their commitment to evolving in line with technological advancements. As organizations continue on their digital transformation journeys, it is important to maintain a focus on sustainability and continually explore new opportunities for improvement. The mission to create a leaner, more modern organization should not overlook the importance of sustainable practices and the responsibilities organizations have towards the environment and society. This entails a continuous commitment to innovation, adaptation, and embracing emerging technologies to drive both digital transformation and sustainability hand in hand. It was clear that, although the interviewees talked about the resources and economics of service levels, they were also occupied with long-term societal sustainability as an important part of sustainability of the future. They had faith in the process and were very optimistic about continuing the city's digital journey. Interviewee 6 did not think he'd heard of anyone who did not see the benefits of a digital journey. Interviewees 2 and 4 agreed that Reykjavík City's digital journey was not over and would take some time. Interviewee 3 agreed and added that it was important for the city of Reykjavík to get out of "...the old shackles of old systems", and added, "... the digital journey and transformation will never end".

*4.4. Innovation in Service Provision*

All interviews have a clear line of thinking that the primary purpose of the city's digital transformation is not just to clean up the paper trail; it is much more than that. It is a new type of thinking, as all processes are transformed primarily to the user first before the system. Hence, digital transformation also carries societal advances, creating better service and citizen experience. Interviewee 13 puts this forth in a transparent manner.

> *It is a fundamental difference in thinking, a "paradigm shift" in how we think about public services. Now, we are centering the service around the user. That is how we create and develop a better city for the future.*

This line of thinking, to put the citizens first, is a breakthrough in providing services as a much broader approach about who needs what, how, and when. Thus, the approach is inclusive, aiming to reach all citizens in all corners of society. While the old approach was exclusive, as people needed cognitive and physical ability to approach many services, the new approach is inclusive and seeks to provide services for all. Interviewee 10 explained the approach clearly: "Individuals have different cognitive and general capacities such as in the physical capacity to move between places…. it is important to make the service approachable".

This change is important as institutions and public institutions especially tend to be very slow to change. Interviewee 6 emphasized that "public institutions tended to be risk averse, the legal framework is much narrower, and rules are difficult to change". Innovation and change management are fundamental to sustainability; the interviewees emphasized how we need new ideas and ways to do things to solve future modern problems. In line with this, interviewee 13 pointed out, "We are creating new paths, changing how we do things, changing the thoughts and the culture… in the end, it will create a measurable difference".

Breaking down silos and top-down thinking within the city's organization is of particular importance, where cooperation is a crucial ingredient. As interviewee 13 framed it, "We must be cooperative; we must engage all departments, managers at all levels". Overall, the interviewees claimed that this shift towards a different organizational chart based on more cooperation and less silo thinking was happening, as interviewee 11 pointed out: "Yes, there is a lot of cooperation, and obviously, this project cannot happen without it… it is essential to unite everyone involved to prevent task duplication". These new approaches and new ways of thinking within the city are evident in the concluding words of interviewee 13:

> *The aim is that the service user feels like they are talking to one person. Hence, we who are behind the scenes need to break down the walls between us to welcome them as a united voice speaking the same language, saying the same thing, and providing the same information because we have collected the information all into one place and opened it up for everyone to access.*

## 5. Discussion and Conclusions

It is recognized that sustainability is a long-term endeavor, and that the role of change management is to foster sustainable practices that are not just short-term projects. They should be integrated into an institution's values and operations. This can involve adopting sustainability to new technologies and innovative practices. This study aimed to gain insight into the experience of Reykjavík city employees in the city's digital transformation on its way to long-term sustainability efforts, coined as the Green Plan. The goal was to shed light on what challenges employees encountered as factors during a large-scale implementation of the Green Plan sustainability efforts. The Green Plan project, an umbrella project for the city of Reykjavík's transformation, is fundamentally an urban sustainability plan that is founded on the UN's sustainable development goals.

The results of the study showed that the interviewees feel a great need for the digital transformation of the city of Reykjavík to go forward with the Green Plan. Technological development is rapidly evolving, increasing users' demands for services in electronic

forms. As Bridges and Bridges [35] point out, organizations face significant changes in their operating environment as new technologies change the nature of communication. At the beginning of change, it is essential that the need for change is clear, and the future vision is convincingly communicated to staff [4,16,19]. In addition to recognizing the need for change, successful change requires stakeholders' involvement, top management's support, and political support for sustainable operations [4,19]. The digital transformation of the city of Reykjavík is part of a more extensive set of digital transformations by the Icelandic state and municipalities. Therefore, it can be considered probable that such political support exists. The interviewees were also aware of the influence of politics and other interested parties on the changes, and one could sense satisfaction in how much effort has been put into the project. Interviewees agreed that the role of managers and leaders is essential in times of change. Those who lead digital transformation projects themselves need to have faith in the projects to encourage other employees. Furthermore, managers need to be able to share information, trust their employees, and motivate them [17–19,36].

Extensive changes take time, and the digital transformation of Reykjavík is a long process, consisting of many projects that contribute to transferring services to users in an electronic form and facilitating the work of employees [16]. The Green Plan is Reykjavík's strategy for a digital journey that lays the foundation for the city's future vision of digital transformation [9]. While vision plays a significant role in Kotter's [16] eight-stage model, stage three consists of developing a vision and strategy, and stage four communicates a new vision. A clear vision was essential to help engage employees in the change initiative [4]. As the city of Reykjavík is a significant workplace, with many employees in many departments and establishments, it can be challenging to get employees to embrace ideas of change. Interviewees discussed the impact of different technical knowledge on users and their struggle to adapt to new technologies, leading to uneven digital participation. Interviewees mentioned the above as an obstacle the city would need to consider carefully and find a solution for. Similarly, Langer et al. [24] have argued that digital equality is essential to ensure all groups can not only help, but also take advantage of the improved services.

The findings demonstrate a connection to all three urban sustainability dimensions. Firstly, although not the main target of digital transformation, the change reduced carbon print with less traffic and reduced the use of paper. Secondly, it increases economic sustainability, as in less time spent on behalf of citizens in accessing services, and more value in time for staff providing and organizing services. The last dimension is social sustainability, which is one of the main arguments for digitalizing services. Social equity and equal access to services for all citizens are two of the prime goals of the Green Plan, as well as the digitalization of the city's services. Although less tangible than the other dimensions, it is of equal importance and is an essential step towards a healthy and sustainable community.

The study conducted in this research highlights the significance of managing change in the public sector to foster sustainability. However, there are still areas that warrant further investigation for future studies. One potential avenue for future research is conducting a comprehensive impact assessment of the digital transformation efforts in the public sector. This assessment can evaluate the effectiveness and efficiency of digital solutions in enhancing service provision, resource management, and overall sustainability. Moreover, it can explore digital development's social, economic, and environmental impacts, providing a more holistic understanding of the benefits and challenges associated with digital transformation in the public sector.

Another critical area for future studies is the examination of the long-term sustainability of digital transformation efforts in the public sector. This research can explore the challenges and opportunities that arise with digital technologies' continuous adaptation and evolution. Strategies to future-proof digital solutions can also be explored, aiming to ensure their relevance and effectiveness in an ever-changing technological landscape. By addressing these research gaps, valuable insights and recommendations can be generated

for policymakers, managers, and practitioners, aiding them in promoting sustainability and enhancing service delivery in the digital era.

**Author Contributions:** Conceptualization, S.G. and T.O.S.; methodology, S.G., E.M.H., I.S.I. and S.D.F.; formal analysis, S.D.F., E.M.H., I.S.I. and S.G.; writing—original draft preparation, S.G. and S.D.F.; writing—review and editing, E.M.H., S.G. and T.O.S. All authors have read and agreed to the published version of the manuscript.

**Funding:** This research received no external funding.

**Institutional Review Board Statement:** Not applicable.

**Informed Consent Statement:** Informed consent was obtained from all subjects involved in the study.

**Data Availability Statement:** The data presented in this study are available upon request from the corresponding author.

**Conflicts of Interest:** The authors declare no conflict of interest.

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
