# Peer review of "Sustainable Digital Change: The Case of a Municipality"

_sustainability, doi:10.3390/su16031319_

Round 1

Reviewer 1 Report

Comments and Suggestions for Authors

Thank you very much for your manuscript, I read it with interest. I have one crucial comment and several major ones. The crucial comment concerns the concept of sustainability and therefor fit for the journal – although the text mentions sustainability and UN sustainable goals several times, I am not convinced that this paper is about sustainability. I think this paper is about administrative changes in a city and I am therefore not fully convinced that Sustainability is a suitable journal for this manuscript. But I am suggesting major revisions so you can present your arguments connected to sustainability clearer in the revised version. Here are other major issues:

·      Section between lines 67 and 92 do not really fit into “Literature review” as it does discuss empirical background and not existing literature.

·      The rest of the “Literature review” is mostly very mechanical (this work said this, this other work said this, etc.) – lines 154-178 are written in a bit more analytical manner, but I would recommend the whole section to be revised to be more analytical and less mechanical.

·      It is also not really clear what is the result of the literature review – what does this section tell you regarding your research?

·      Section 2.2 is supposed to be part of “Literature review”, but it is – again – a combination of background empirical information and existing literature. This mixture provides a rather unclear results and we do not know what this part (2.2) brings to the table from the perspective of your analysis.

·      The paper does not have an explicit research question – please add it to your text.

·      Methodology section is not clearly structured, for example, it walks about grounded theory on two separate places.

·      Although the authors claim that they do grounded theory, they do not really do any theory/method at all (some consider grounded theory to be method, others to be theory) – they do not build any theory, they just describe their data. This is OK with me, just do not call it grounded theory when it is not.

·      The analytical part has four section (although there are two sections 4.3), but it is not clear here these created.

·      Although section 4.3 is called “Sustainability in resources” I am not sure if this is what the section is about. Large majority of excerpts from interviews discuss convenience of starting to do things electronically. To be perfectly honest, the only thing I can consider an issue of sustainability is less travelling needed identified in some excerpts – but these are not interpreted in sustainable way by the authors.

Author Response

Dear Reviewer,

Thank you for your valuable feedback. We appreciate your constructive comments on our manuscript. We have carefully considered your suggestions and made significant revisions to address the concerns you raised. In attachment you will find our response to each of your comments, along with the corresponding changes made to the manuscript.

Reviewer 2 Report

Comments and Suggestions for Authors

Dear Author(s),

The presented article on digital change in public services has a clear idea, an appropriate method and is well organized. The results of the survey can contribute to improving sustainable digital changes in public institutions in the municipalities of Reykjavík. Although the author(s) take a sound managerial view of digital transformation in this area, it would also be useful for future research to consider additional viewpoints to compare current digital changes with others that have occurred in the past, to find their distinctive characteristics for management and also taking into account attention to wide international experience.

Author Response

Dear Reviewer,

Thank you for your valuable feedback. We appreciate your constructive comments on our manuscript. We have carefully considered the feedback and made significant revisions to strengthen the focus and alignment of the study with the concept of sustainability, as outlined in the "Call for Papers" of Sustainability. The Literature Review has been refined to create a more cohesive and clear research question. Additionally, the authors have highlighted the need to include diverse perspectives when discussing the implementation of digital changes, taking into account the wide range of international experiences in this area. The suggestion to address sustainability initiatives on a larger scale is appreciated and has been incorporated into the revised version of the manuscript. Thank you once again for the valuable input.

Reviewer 3 Report

Comments and Suggestions for Authors

After Covid 19, the global world has moved to the BANI model. The impact of this model on the sustainability of digital change should be added.

  This is a fundamental global shift that will require scholars to expand the list of literature and adapt the model and methods.

Comments on the Quality of English Language

The paper needs to be edited.

Author Response

Dear Reviewer, thank you for your comment on the impact of the BANI (Brittle, Anxious, Nonlinear, Incomprehensible) model on the sustainability of digital change. While the BANI model is indeed relevant in the context of post-Covid global shifts and digital transformations, the focus of our study is primarily on the management of change and sustainability in the public sector. We acknowledge that exploring the intersection between the BANI model and sustainability in future research could greatly contribute to the literature. However, due to the scope and objectives of our current study, we have not specifically addressed the BANI model in this manuscript. We appreciate your suggestion, and we agree that further research in this area can expand on the model and methods applied in light of the evolving global landscape.

Reviewer 4 Report

Comments and Suggestions for Authors

The purpose of this manuscript is to evaluate the experience of Reykjavík city employees in the city's digital transformation leading to a sustainable operation.
The topic is interesting. The results of the analysis enrich the research in this field. However, from my point of view, a number of revisions are needed.
First, the authors must present clearer arguments that the research is related to sustainability. Furthermore, they must develop the conclusion section adding the impact of this model on the sustainability of digital changeand clearly mention to whom the obtained results are useful.
Secondly, they should identify other current bibliographic sources and report their results to these sources (Almost half of the bibliographic sources used by the authors are published in the early 2000s). Also, when reviewing the literature, the analysis should be a little more analytical and ensure the connection between the analysed literature and the present research.
Third, authors must clearly state which theoretical framework they used.

Overall, I evaluate the study very positively and I recommend its publication after minor revisions.

Author Response

Dear Reviewer.

Thank you for your valuable feedback. We appreciate your constructive comments on our manuscript. We have carefully considered your suggestions and made significant revisions to address the concerns you raised. In the attached document we provide a response to each of your comments, along with the corresponding changes made to the manuscript.

Round 2

Reviewer 1 Report

Comments and Suggestions for Authors

Dear Authors, 

thank you very much for your revisions. I think the paper is much clearer and much more connected to the issue of sustainability. I have only one outstanding issue - I was not able to find your answer to my last point that concerns section 4.3 and the section has not been revised (there are some very small text revisions, but nothing that would reflect on my point. Please revise the text or provide rebuttal. 

This is the comment from my previous review:

·"Although section 4.3 is called “Sustainability in resources” I am not sure if this is what the section is about. Large majority of excerpts from interviews discuss convenience of starting to do things electronically. To be perfectly honest, the only thing I can consider an issue of sustainability is less travelling needed identified in some excerpts – but these are not interpreted in sustainable way by the authors."

Sincerely

Reviewer

Author Response

We appreciate your input and recognize the potential to align this section more closely with the central objective of the paper. To achieve this, we have made revisions to enhance clarity and reimagined the section's title to better reflect its content and alignment with the paper's overarching goal. Furthermore, we have enriched the findings by incorporating additional insights from the interviews, thereby providing a more comprehensive coverage of the topic

Reviewer 3 Report

Comments and Suggestions for Authors

I agree with authors' argue about BANI

Comments on the Quality of English Language

Is OK

Author Response

The manuscript has now been proofread.